# REVISITING KNOWLEDGE BASE EMBEDDING AS TENSOR DECOMPOSITION

## ABSTRACT

We study the problem of knowledge base (KB) embedding, which is usually addressed through two frameworks—neural KB embedding and tensor decomposition. In this work, we theoretically analyze the neural embedding framework and subsequently connect it with tensor based embedding. Specifically, we show that in neural KB embedding the two commonly adopted optimization solutions—margin-based and negative sampling losses—are closely related to each other. We also reach the closed-form tensor that is implicitly approximated by popular neural KB approaches, revealing the underlying connection between neural and tensor based KB embedding models. Grounded in the theoretical results, we further present a tensor decomposition based framework KBTD to directly approximate the derived closed form tensor. Under this framework, the neural KB embedding models, such as NTN (Socher et al., 2013), TransE (Bordes et al., 2013), Bilinear (Jenatton et al., 2012), and DISTMULT (Yang et al., 2015), are unified into a general tensor optimization architecture. Finally, we conduct experiments on the link prediction task in WordNet and Freebase, empirically demonstrating the effectiveness of the KBTD framework.

## 1 INTRODUCTION

Knowledge bases (KBs) power many of semantic-oriented techniques and applications, such as question answering and intelligent personal assistant. A classical example is the automatic answer to the query "Who is Barack Obama's wife" by KB-supported search engines. Most if not all of KBs achieve this by storing the facts about the world in the form of RDF triplets (W3C, 1999), wherein a triplet (*subject, predicate, object*), in short (*s, r, o*), records a piece of fact about the relation between the two entities—the *subject* and *object*. To automatically construct Web-scale KBs with billions of facts (triplets), a significant line of effort has been devoted to knowledge base embedding—the technique of encoding entities and their relational information into latent representations (Bordes et al., 2011; 2013; Socher et al., 2013; Yang et al., 2015).

In particular, the direction of neural embedding has been extensively explored for learning representations for KBs, offering state-of-the-art performance for validating and completing unseen facts (Yang et al., 2015). Briefly, given a KB represented by triplets $T = \{(s, r, o)\}$, neural embedding models take these known triplets as positive instances and corrupted triplets $\{(s', r, o')\}$ as negative ones. For each triplet, a scoring function $f(s, r, o)$— parameterized with a neural network—is designed to project the associated entities $s, o$ and their relational information $r$ into a scalar. Most of the existing models are then trained through two popular choices of loss functions, including the margin-based ranking loss by NTN (Socher et al., 2013), TransE (Bordes et al., 2013), and DISTMULT (Yang et al., 2015), as well as several practices of the Negative Sampling loss (Mikolov et al., 2013) by Bilinear (or LFM) (Jenatton et al., 2012) and CONV (Toutanova et al., 2015). In addition, another line of KB embedding is focused on tensor decomposition based frameworks, such as RESCAL (Nickel et al., 2011; 2012).

Notwithstanding the rapid development and progress of KB embedding techniques, insights concerning their underlying mechanisms are to date sorely lacking. For example, a natural question that arises is what are the relationships or differences between the margin-based ranking loss and the negative sampling loss for neural KB embedding. Moreover, what is the quantity that is optimized by conventional neural KB embedding models? With an eye toward comprehensively understanding

Table 1: Knowledge Base Embedding.

| local margin-based loss (ReLU) | $\max\left(0, f(s', r, o') - f(s, r, o) + \gamma\right)$ |
|---|---|
| local margin-based loss (Softplus) | $\log\left(1 + e^{f(s', r, o') - f(s, r, o) + \gamma}\right)$ |
| local negative sampling loss | $\log\left(1 + e^{f(s', r, o') - f(s, r, o)} + e^{f(s', r, o')} + e^{-f(s, r, o)}\right)$ |
| closed-form tensor | $\log\left(\frac{2|E|\mathcal{X}_{s,r,o}}{b \cdot \left(d_{s,r}^{\text{out}} + d_{o,r}^{\text{in}}\right)}\right)$ |

See detailed notations in Sections 3 and 4. A brief introduction is listed below:

- $(s, r, o)$ & $(s', r, o')$: the known and corrupted triplets, respectively;
- $E$ & $R$: the entity and relation sets of a given knowledge base, with $s, o \in E$ and $r \in R$;
- $\mathcal{X} \in \{0, 1\}^{E \times R \times E}$: a three-way binary tensor, with $\mathcal{X}_{s,r,o} = 1$ indicating $(s, r, o) \in T$ and otherwise 0;
- $d_{s,r}^{\text{out}} = \sum_o \mathcal{X}_{s,r,o}$ & $d_{o,r}^{\text{in}} = \sum_s \mathcal{X}_{s,r,o}$: the out-/in- degree of entity $s$/$o$ under relation $r$, respectively;
- $b$: the number of negative samples.

KB embedding, we investigate (1) the connection between margin-based ranking loss and negative sampling loss in neural KB models, (2) the relationship between neural KB models and classical tensor-based KB models, and (3) the universal framework for KB learning.

**Contributions.** In this work, we unveil two fundamentals of KB embedding, according to which we further present a tensor decomposition based KB embedding framework—KBTD, yielding significant outperformance over neural KB embedding models in most cases.

First, with softplus' smooth approximation to ReLU in the margin-based loss (Dugas et al., 2001), we show that the margin-based loss is closely connected to the negative sampling loss (See rows 2 & 3 in Table 1). In specific, both losses aim to encourage positive triplets $(s, r, o)$ and penalize corrupted ones $(s', r, o')$, and the slight difference lies in the extra reward or penalization to corresponding triplets in the negative sampling loss.

Second, we derive the closed form tensor (See row 4 in Table 1) whose entry is implicitly fitted (approximated) by scoring function $f(s, r, o)$, when optimizing neural KB embedding models through the negative sampling loss. This closed form generalizes the ultimate objective of previous attempts on designing various scoring functions, such as NTN, TransE, Bilinear, and DISTMULT. This finding also links the neural KB embedding framework with the tensor-based KB embedding approach.

Third, building upon the discoveries, we propose a tensor decomposition based KB embedding framework, KBTD, to directly fit the closed form tensor with by leveraging the scoring functions proposed in several popular neural models. Our extensive experiments on WordNet and FreeBase demonstrate the outstanding performance of KBTD over the conventional margin-based neural framework. In addition, we point out the limitation of dissimilarity/distance based scoring function design, which is wildly adopted by the TransE/H/R/D models (Bordes et al., 2013; Wang et al., 2014; Lin et al., 2015; Ji et al., 2015).

The rest of this paper is organized as follows. Section 2 discusses related work. Section 3 unveils the connection between the margin-based ranking loss and the negative sampling loss. Section 4 performs the theoretical analysis and subsequently presents our KBTD framework. Section 5 introduces the detailed experiments on the link prediction task for KBs. Section 6 concludes this paper.

## 2 RELATED WORK

Knowledge base embedding is being extensively explored and developed over the last few years, during which the major breakthroughs are resulted from the neural embedding and tensor factorization models (Bordes et al., 2013; Nickel et al., 2012). Our work focuses on understanding the fundamentals of neural KB embedding, as well as its connection with tensor models.

**Neural KB Embedding: Loss Functions.** Neural knowledge base embedding is usually formulated as an optimization problem with different loss functions. The majority of existing KB models employ the margin-based ranking loss, which was first proposed (Collobert et al., 2011) for addressing the efficiency issue of softmax (Bengio et al., 2003) in the field of natural language models. A brief collection of recent margin-based KB embedding methods include the SE, Unstructured, and SME models (Bordes et al., 2011; 2012; 2014), SLM and NTN (Socher et al., 2013), DISTMULT (Yang

et al., 2015), TransE (Bordes et al., 2013), TransH (Wang et al., 2014), TransR (Lin et al., 2015), TransD (Ji et al., 2015), and ProjE (Shi & Weninger, 2017). In addition, there are several models— Bilinear model (Jenatton et al., 2012) and CONV (Toutanova et al., 2015)—that adopt the negative sampling (NE) loss. Our work contributes to this line of research by providing the relationship between margin-based and negative sampling based KB embedding models.

**Neural KB Embedding: Scoring Functions.** As summarized in (Yang et al., 2015), given a KB represented by a list of triplets, neural models learn embeddings by utilizing a neural network, wherein the first layer projects the two entities of each triplet into latent low-dimensional vectors, and the second layer leverages a scoring function to operate on each pair of entity vectors with relation-specific parameters. The major difference between neural KB models lie in the various ways that they design the scoring functions — TransE, NTN, Bilinear, and DISTMULT (See details in Table 3 of Section 5). To date, few attempts have been conducted to understand any common grounds behind these models. Our work furthers this direction by proposing a general tensor decomposition framework (KBTD) which unifies existing neural KB models.

**Tensor Decomposition for KB Embedding.** Tensor decomposition has seen successes in structural and relational learning over decades (Kolda & Bader, 2009; Sun et al., 2006). Recent years also witness the natural application of this technique on learning KB embeddings, including the BCTF model (Sutskever et al., 2009), and RESCAL (Nickel et al., 2011). In this work, we show the closed relationship between tensor decomposition based KB embedding and neural KB embedding models.

# 3 CONNECTING MARGIN-BASED LOSS WITH NEGATIVE SAMPLING LOSS

Given a KB with entity set $E$ and relation set $R$, represented by a set of triplets $T = \{(s, r, o)\}$ with $s, o \in E$ and $r \in R$, the goal of neural KB models is to learn a scoring function $f(s, r, o)$ which evaluates an arbitrary triplet and outputs a scalar to measure the acceptability of this triplet, where high/low score indicates that the input triplet tends to be correct/wrong. As summarized by (Yang et al., 2015), most existing scoring functions can be unified by a neural network, where the first layer projects the two entities of each triplet into latent low-dimensional vectors, and the second layer applies either linear or bilinear transformation (or both) on entity vectors with relation-specific parameters.

The scoring function is then fitted to a loss function to learn the representations of both entities and relations. The majority of neural KB embedding models adopt either a margin-based ranking loss or a negative sampling loss. Both loss functions leverage the known triplets $T$ as positive samples and the corrupted triplets $T'$ as the negative ones. Following the literature, given a known triplet $(s, r, o) \in T$, its corrupted triplets $T'_{(s,r,o)}$ are constructed by replacing either the subject entity $s$ or the object entity $t$ with an arbitrary entity from $E$, i.e., $T'_{(s,r,o)} = \{(s', r, o)|s' \in E\} \cup \{(s, r, o')|o' \in E\}$. Given both positive and corrupted triplets, the objective of the margin-based ranking loss is to minimize:

$$\mathcal{L}_{\text{MARGIN}} = \sum_{(s,r,o) \in T} \sum_{(s',r,o') \in T'_{(s,r,o)}} \max\big(0, \gamma + f(s', r, o') - f(s, r, o)\big).$$

Its challenge lies in, however, the second summation, which takes $O(|E|)$ complexity to enumerate the whole entity set $E$ and is extremely time demanding. Therefore, in practice, the second summation is commonly approximated with sampling and the sample size is usually set to one (Bordes et al., 2013). Considering the local loss function $\ell_{\text{MARGIN}}(s, r, o)$ for each positive triplet $(s, r, o)$ associated with one (sampled) corrupted triplet $(s', r, o')$, the $\max(0, \cdot)$ loss can be smoothly approximated by the *softplus* function (Dugas et al., 2001), that is:

$$\begin{aligned}
\ell_{\text{MARGIN}}(s, r, o) &= \max\big(0, \gamma + f(s', r, o') - f(s, r, o)\big) \\
&\approx \log\big(1 + e^{\gamma + f(s',r,o') - f(s,r,o)}\big).
\end{aligned} \tag{1}$$

On the other hand, the negative sampling loss aims to optimize the following objective:

$$\mathcal{L}_{\text{NEG}} = - \sum_{(s,r,o) \in T} \Big(\log \sigma(f(s, r, o)) + b\mathbb{E}_{(s',r,o') \sim T'_{(s,r,o)}} \big[\log \sigma(-f(s', r, o'))\big]\Big), \tag{2}$$

where $b$ is the number of negative samples and $\sigma(\cdot)$ is the sigmoid function. Similarly, the expectation term can be replaced with its Monte Carlo approximation. To align with our previous

discussion on the margin-based ranking loss, we also set negative sample size $b = 1$ and derive the local objective for a certain positive triplet $(s, r, o)$ to be:

$$
\begin{aligned}
\ell_{\text{NEG}}(s, r, o) &= -\log \sigma(f(s, r, o)) - \log \sigma(-f(s', r, o')) \\
&= \log \left(1 + e^{f(s', r, o') - f(s, r, o)} + e^{f(s', r, o')} + e^{-f(s, r, o)}\right).
\end{aligned}
\tag{3}
$$

Eq. 1 and Eq. 3 reveal the close relationship between margin-based ranking loss and negative sampling loss in neural KB embedding. First, observed from the similar term $e^{f(s', r, o') - f(s, r, o)}$, both loss functions implicitly encourage positive triplets to have relatively higher scores than the corrupted ones. Second, the extra term $e^{f(s', r, o')} + e^{-f(s, r, o)}$ in the negative sampling loss suggests that in addition to the implicit comparison, it explicitly rewards the positive triplets to have high scores, and also encourages the corrupted ones to have low scores.

## 4    Unifying Neural KB Embedding as Tensor Decomposition

From the above section, we observe that the margin-based loss and negative sampling loss share very similar form. In this section, we unify existing neural KB embedding models by assuming a general scoring function $f : E \times R \times E \to \mathbb{R}$ and the use of negative sampling loss. We present a theoretical analysis in Section 4.1, followed by our KBTD framework which formally defines KB embedding problem as a tensor decomposition problem in Section 4.2. Additionally, the connection between KBTD and classical tensor decomposition models (Nickel et al., 2011; 2012) is discussed in Section 4.2.

### 4.1    Theoretical Analysis of Neural KB Embedding

To facilitate our analysis, we represent the KB triplets as a three-way binary tensor $\mathcal{X} \in \{0, 1\}^{E \times R \times E}$, where $\mathcal{X}_{s,r,o} = 1$ indicates $(s, r, o) \in T$, while $\mathcal{X}_{s,r,o} = 0$ for non-existing or unknown triplets. The loss function $\mathcal{L}_{\text{NEG}}$ in Eq. 2 can be re-formatted with $\mathcal{X}_{s,r,o}$ as

$$
-\sum_{s,r,o} \mathcal{X}_{s,r,o} \left\{ \log \sigma\left(f(s, r, o)\right) + \frac{b}{2} \mathbb{E}_{o' \sim P_N} \left[\log \sigma\left(-f(s, r, o')\right)\right] + \frac{b}{2} \mathbb{E}_{s' \sim P_N} \left[\log \sigma\left(-f(s', r, o)\right)\right] \right\},
$$

where $P_N$ is a uniform distribution over all entities, i.e., $P_N(\cdot) = \frac{1}{|E|}$. We further break down the summation and arrive at the following form:

$$
\begin{aligned}
\mathcal{L}_{\text{NEG}} = &-\sum_{s,r,o} \mathcal{X}_{s,r,o} \log \sigma\left(f(s, r, o)\right) \\
&- \frac{b}{2} \left( \sum_{s,r} d_{s,r}^{\text{out}} \mathbb{E}_{o' \sim P_N} \left[\log \sigma\left(-f(s, r, o')\right)\right] + \sum_{r,o} d_{o,r}^{\text{in}} \mathbb{E}_{s' \sim P_N} \left[\log \sigma\left(-f(s', r, o)\right)\right] \right),
\end{aligned}
\tag{4}
$$

where $d_{s,r}^{\text{out}} = \sum_o \mathcal{X}_{s,r,o}$ is the out-degree of entity $s$ under relation $r$, and $d_{o,r}^{\text{in}} = \sum_s \mathcal{X}_{s,r,o}$ is the in-degree of entity $o$ under relation $r$. Then we explicitly express the two expectation terms:

$$
\begin{aligned}
\mathbb{E}_{o' \sim P_N} \left[\log \sigma\left(-f(s, r, o')\right)\right] &= \sum_{o'} \frac{1}{|E|} \log \sigma\left(-f(s, r, o')\right) \\
&= \frac{1}{|E|} \log \sigma\left(-f(s, r, o)\right) + \sum_{o' \neq o} \frac{1}{|E|} \log \sigma\left(-f(s, r, o')\right) \\
\mathbb{E}_{s' \sim P_N} \left[\log \sigma\left(-f(s', r, o)\right)\right] &= \frac{1}{|E|} \log \sigma\left(-f(s, r, o)\right) + \sum_{s' \neq s} \frac{1}{|E|} \log \sigma\left(-f(s', r, o)\right).
\end{aligned}
$$

Then, by utilizing the above expectation terms, the local loss function for each specific triplet $(s, r, o)$ in Eq. 4 can be defined as

$$
\ell(s, r, o) = -\mathcal{X}_{s,r,o} \log \sigma\left(f(s, r, o)\right) - \frac{b \cdot \left(d_{s,r}^{\text{out}} + d_{o,r}^{\text{in}}\right)}{2 |E|} \log \sigma\left(-f(s, r, o)\right).
$$

Table 2: The comparison between RESCAL and our KBTD framework with parameters $\Theta$ initialized in the bilinear way, that is, $\Theta = \{\boldsymbol{a}_1, \cdots, \boldsymbol{a}_{|E|}, \boldsymbol{W}_1, \cdots, \boldsymbol{W}_{|R|}\}$.

| RESCAL | $\min_\Theta \sum_{s,r,o} \left(\mathcal{X}_{s,r,o} - \boldsymbol{a}_s^\top \boldsymbol{W}_r \boldsymbol{a}_t\right)^2$ |
|---|---|
| KBTD | $\min_\Theta \sum_{s,r,o} \mathcal{W}_{s,r,o} \left(\mathcal{Y}_{s,r,o} - \boldsymbol{a}_s^\top \boldsymbol{W}_r \boldsymbol{a}_t\right)^2$ |

The work in (Levy & Goldberg, 2014) suggested that for sufficient large embedding dimensionality, each individual $f(s,r,o)$ can assume a value independence[1]. Following this assumption enables us to treat the objective $\mathcal{L}$ as a function of independent $f(s,r,o)$ terms. The partial derivative with respect to $f(s,r,o)$ can be taken as:

$$\frac{\partial \mathcal{L}}{\partial f(s,r,o)} = \frac{\partial \ell(s,r,o)}{\partial f(s,r,o)} = -\mathcal{X}_{s,r,o}\sigma\left(-f(s,r,o)\right) + \frac{b \cdot \left(d_{s,r}^{\text{out}} + d_{o,r}^{\text{in}}\right)}{|E|}\sigma\left(f(s,r,o)\right).$$

By setting the derivative to zero, we have

$$e^{2f(s,r,o)} - \left(\frac{2\,|E|\,\mathcal{X}_{s,r,o}}{b \cdot \left(d_{s,r}^{\text{out}} + d_{o,r}^{\text{in}}\right)} - 1\right)e^{f(s,r,o)} - \frac{2\,|E|\,\mathcal{X}_{s,r,o}}{b \cdot \left(d_{s,r}^{\text{out}} + d_{o,r}^{\text{in}}\right)} = 0,$$

which implies

$$f(s,r,o) = \log\left(\frac{2\,|E|\,\mathcal{X}_{s,r,o}}{b \cdot \left(d_{s,r}^{\text{out}} + d_{o,r}^{\text{in}}\right)}\right). \tag{5}$$

The RHS of Eq. 5 defines a "transformed" tensor based on $\mathcal{X}$, and the LHS of Eq. 5 implies a regression problem, i.e., try to fit the $(s,r,o)$-entry of the transformed tensor with the scoring function $f(s,r,o)$. In next section, we formally define this problem and then propose our KBTD framework.

## 4.2 KBTD: NEURAL KB EMBEDDING AS TENSOR DECOMPOSITION

In this section, we formalize the KB embedding problem analyzed in Section 4.1 as a tensor decomposition problem. We further present our framework—KBTD—to learn latent embedding for KB entities and relations. Its connection with classical tensor decomposition methods is also discussed.

First, as mentioned at the end of Section 4.1, RHS of Eq. 5 defines a transformed tensor based on $\mathcal{X}$. Here we denote it to be tensor $\mathcal{Y} \in \mathbb{R}^{|E| \times |R| \times |E|}$, with the $(s,r,o)$-entry defined to be

$$\mathcal{Y}_{s,r,o} = \log\left(\frac{2\,|E|\,\mathcal{X}_{s,r,o}}{b \cdot \left(d_{s,r}^{\text{out}} + d_{o,r}^{\text{in}}\right)}\right). \tag{6}$$

Second, our discussion in Section 4.1 actually implies a weighted tensor decomposition problem. This is mainly due to the negative sampling mechanism — we only care about positive triplets and corrupted triplets. This mechanism can be characterized by a binary tensor $\mathcal{W} \in \{0,1\}^{|E| \times |R| \times |E|}$, wherein $\mathcal{W}_{s,r,o} = 1$ if and only if $(s,r,o)$ is either a positive triplet or a corrupted triplet. Given the definition of tensor $\mathcal{Y}$ and tensor $\mathcal{W}$, we can formalize the following tensor decomposition problem:

$$\min_\Theta \sum_{s,r,o} \mathcal{W}_{s,r,o} \left(\mathcal{Y}_{s,r,o} - f_\Theta(s,r,o)\right)^2, \tag{7}$$

where $f_\Theta$ is the scoring function parameterized by $\Theta$.

**Revisiting RESCAL (Nickel et al., 2011; 2012).** Before introducing how we optimize Eq. 7, we would like to discuss the connection between KBTD and RESCAL—a classical tensor decomposition model for KB embedding. Table 2 lists the optimization problems solved by RESCAL and our framework, in which the scoring function in Eq. 7 is initialized as a bilinear function, i.e., $f(s,r,o) = \boldsymbol{a}_s^\top \boldsymbol{W}_r \boldsymbol{a}_o$, where $\boldsymbol{a}_s, \boldsymbol{a}_o \in \mathbb{R}^d$ and $\boldsymbol{W}_r \in \mathbb{R}^{d \times d}$. We observe the following connections and differences between them. First, both models explain a RDF triplet $(s,r,o)$ through the latent

---

[1]We realize that there has been discussion that this supposition may not be rigorous enough (Arora et al., 2016).

---

**Algorithm 1:** The KBTD Framework

---

**input:** Training set $T = \{(s, r, o)\}$, entity and relation set $E$ and $R$, corrupted triplets multiplier $\lambda$, mini-batch size $B$

**output:** Models parameters $\Theta$, including entity and relation embeddings

1 Initialize model parameters $\Theta$;
2 **while do**
    /* Sample a mini-batch of size $B$                                  */
3     $T_{batch} \leftarrow \text{sample}(T, B)$;
    /* Sample corrupted triplets for this mini-batch       */
4     $T'_{batch} \leftarrow \emptyset$;
5     **for** $(s, r, o) \in T_{batch}$ **do**
6         **for** $i = 1$ **to** $\lambda$ **do**
7             $s' \leftarrow \text{sample}(E)$;
8             $o' \leftarrow \text{sample}(E)$;
9             $T'_{batch} \leftarrow T'_{batch} \cup (s', r, o) \cup (s, r, o')$ ;
10     Update parameter $\Theta$ w.r.t. $\sum_{(s,r,o) \in T_{batch} \cup T'_{batch}} \left( \mathcal{Y}_{s,r,o} - f_\Theta(s, r, o) \right)^2$;

---

representations $\boldsymbol{a}_s, \boldsymbol{a}_o$ and $\boldsymbol{W}_r$. To learn the representations, however, RESCAL directly factorizes the binary tensor $\mathcal{X}$, while our model decomposes a transformed real-value tensor $\mathcal{Y}$. Second, KBTD also differs with RESCAL in the way they treat the unobserved triplets. Notice that given a KB of observed (positive) triplets, the unobserved triples includes both positive and negative ones. This issue is known as the one-class problem (Moya & Hush, 1996; Pan et al., 2008). Two common solutions to this problem are AMAN (all missing as negative) and AMAU (all missing as unknown). The RESCAL model simply adopts the AMAN strategy by assuming all unobserved triplets as negative ones. However, our model is able to implicitly compromise between AMAN and AMAU by only treating corrupted triplets as negative.

**KBTD Learning.** The detailed optimization procedure for KBTD is described in Algorithm 1. We optimize the objective function in Eq. 7 using mini-batch stochastic gradient descent with Ada-Grad (Duchi et al., 2011). At each main iteration (Line 3-10), we first sample a mini-batch of positive triplets (Line 4) and then sample their corrupted triplets whose size is controlled by a multiplier $\lambda$ (Line 6-10). The parameters are updated with respect to the sampled positive triplets as well as corresponding corrupted ones. In this setting, we avoid generating the dense tensors $\mathcal{Y}$ and $\mathcal{W}$, which may in practice result in memory issues.

There is a computational issue that comes from the $\log$ operator. For an unobserved triplet $(s, r, o)$ (i.e., $\mathcal{X}_{s,r,o} = 0$), $\mathcal{Y}_{s,r,o} = \log 0 = -\infty$. Previously, two approaches have been proposed for addressing it (Levy & Goldberg, 2014). One is to smooth the logarithm by adding a small constant to tensor $\mathcal{X}$, generating a dense tensor. The other one is to apply an additional shifted-truncated operator, that is, $\max(\mathcal{Y}_{s,r,o} - c, 0)$, generating a sparse tensor with the loss of certain information. Due to the obvious drawbacks, we instead propose to use a simple and effective solution, wherein the operation $\log x$ is replaced with $\log(\epsilon + x)$ with $\epsilon$ as a tunable parameter.

## 5 EXPERIMENTS

In this section, we evaluate the proposed KBTD framework on the canonical link prediction task against several popular KB embedding methods on two datasets extracted from WordNet and Free-Base. In this task, we are given a KB with a certain fraction of triplets removed, and our target is to predict these missing triplets. We first introduce our experimental setup in Section 5.1, followed by detailed discussion on experimental results in Section 5.2.

### 5.1 EXPERIMENTAL SETUP

**Datasets.** We use WordNet (WN18) and FreeBase (FB15k) datasets as introduced in (Bordes et al., 2013) where WN18 consists of 151, 442 triplets with 40,943 entities and 18 relations, and FB15k contains 592,213 triplets with 14,951 entities and 1,345 relations. We use the same training/validation/test split as in (Bordes et al., 2013; Yang et al., 2015).

Table 3: Scoring Functions and Parameters.

| Model Name | Parameters $\Theta$ | Scoring Function $f_\Theta(s, r, o)$ |
|---|---|---|
| NTN | $\left\{ \boldsymbol{u}_{1,\cdots,|R|}, \boldsymbol{W}_{1,\cdots,|R|}^{[1:k]}, \boldsymbol{V}_{1,\cdots,|R|}, \boldsymbol{a}_{1,\cdots,|E|} \right\}$ | $\boldsymbol{u}_r^\top \tanh\left( \boldsymbol{a}_s \boldsymbol{W}_r^{[1:k]} \boldsymbol{a}_o + \boldsymbol{V}_r \begin{bmatrix} \boldsymbol{a}_s \\ \boldsymbol{a}_o \end{bmatrix} + \boldsymbol{b}_r \right)$ |
| TransE | $\left\{ \boldsymbol{w}_{1,\cdots,|R|}, \boldsymbol{a}_{1,\cdots,|E|} \right\}$ | $-\|\boldsymbol{a}_s + \boldsymbol{w}_r - \boldsymbol{a}_o\|_2^2$ |
| Bilinear | $\left\{ \boldsymbol{W}_{1,\cdots,|R|}, \boldsymbol{a}_{1,\cdots,|E|} \right\}$ | $\boldsymbol{a}_s^\top \boldsymbol{W}_r \boldsymbol{a}_o$ |
| DISTMULT | $\left\{ \boldsymbol{w}_{1,\cdots,|R|}, \boldsymbol{a}_{1,\cdots,|E|} \right\}$ | $\boldsymbol{a}_s^\top \mathrm{diag}(\boldsymbol{w}_r) \boldsymbol{a}_o$ |

Table 4: Experimental Results on the WN18 Dataset.

| | Results from Neural KB Embedding | | Results from our KBTD framework | |
|---|---|---|---|---|
| | MRR | HITS@10 | MRR | HITS@10 |
| NTN | 0.53 | 66.10 | **0.85** | **90.50** |
| TransE | 0.38 | **90.90** | **0.39** | 82.18 |
| Bilinear | 0.89 | 92.80 | **0.92** | **94.62** |
| DISTMULT | **0.83** | 94.20 | 0.81 | **94.62** |

Table 5: Experimental Results on the FB15k Dataset.

| | Results from Neural KB Embedding | | Results from our KBTD framework | |
|---|---|---|---|---|
| | MRR | HITS@10 | MRR | HITS@10 |
| NTN | 0.25 | 41.40 | **0.37** | **59.06** |
| TransE | **0.31** | **53.90** | 0.30 | 49.94 |
| Bilinear | 0.31 | 51.90 | **0.31** | **54.95** |
| DISTMULT | 0.35 | 57.70 | **0.35** | **59.91** |

**Baselines.** We compare our proposed framework with TransE (Bordes et al., 2013), NTN (Socher et al., 2013), Bilinear (Jenatton et al., 2012) and DISTMULT (Yang et al., 2015). The original TransE model is based on dissimilarity/distance function. To fit our framework, we define the scoring function for TransE to be negative dissimilarity/distance function, i.e., $f(s, r, o) = -\|\boldsymbol{a}_s + \boldsymbol{a}_r - \boldsymbol{a}_o\|_2^2$. For NTN, Bilinear and DISTMULT, we inherit the scoring functions from their paper. The detailed scoring functions as well as their parameters are listed in Table 3. For the meaning of parameters and the intuition behind scoring functions, readers can refer to the original papers.

**Evaluation Protocol.** We exactly follow the experimental procedure and treatment used in TransE (Bordes et al., 2013) and DISTMULT (Yang et al., 2015). For each triplet $(s, r, o)$ in the test set, the subject entity $s$ is replaced with each of entities from entity set $E$ in turn. We apply corresponding scoring function $f$ on those corrupted triplets and then sort them in non-increasing order to get the rank of the correct triplet. This procedure is then repeated for the object entity $o$. For evaluation metrics, we consider *Mean Reciprocal Rank (MRR)* which is defined to be an average of the reciprocal rank of the correct triplets over all test triplets, and *HITS@10* (top-10 accuracy). If possible, we list the experimental results reported in (Yang et al., 2015) directly. In addition, we apply the *filtered* setting from (Bordes et al., 2013; Yang et al., 2015) in evaluation. In this setting, for one certain test triplet $(s, r, o)$, we removed from the list of corrupted triplets all the triplets which appear in training, validation, or test set, except $(s, r, o)$ itself. This setting, for example, can avoid cases where lots of triplets in training set rank above the one of interest.

**Implementation Details.** All the models in our framework were implemented using PyTorch in a machine with one 12GB GPU. Since the complexities of the aforementioned approaches vary a lot, in order to achieve the best accuracy for all the models, we cross-validate using the validation set to find the best hyperparameters. We found that, except for TransE on WN18, all the methods on both datasets can share the same hyper-parameters: (1) dimensionality $d = 100$; (2) smoothing parameter $\epsilon = 0.01$; (3) multiplier $\lambda$ mentioned in Algorithm 1 was set to 2; (4) the learning rate of AdaGrad algorithm was set to 0.1 (0.01 for TransE on WN18); (5) $\ell_2$-regularization applied to all the parameters using the weight 0.0001; (6) the mini-batch size is set to 2,048 (4,800 for TransE on WN18); (6) $b = 1$ in Eq. 6 (200 for TransE on WN18). For the additional hyper-parameter in NTN method, i.e., the number of slices $k$, was set to 2. We allow all the algorithms to run at most 10,000 epochs over the training data, and the best model was selected by early stopping using HITS@10

Table 6: Model Complexity in terms of #Parameters. $d$ is the embedding dimension, and $k$ in NTN is the number of slices.

| Methods | # Parameters |
|---------|--------------|
| NTN | $O(|R|\, d^2 k + |E|\, d)$ |
| TransE | $O(|R|\, d + |E|\, d)$ |
| Bilinear | $O(|R|\, d^2 + |E|\, d)$ |
| DISTMULT | $O(|R|\, d + |E|\, d)$ |

score on the validation sets. By taking advantages of GPU computation, every training experiment can be finished within 4 hours.

## 5.2 EXPERIMENTAL RESULTS

Table 4 and Table 5 list the overall results on the WN18 and FB15k datasets for several popular models under both our KBTD framework and the neural KB embedding framework, respectively. In general, we have the following key observations and insights:

(1) On the WN18 dataset, the KBTD framework achieves the best performance among most cases. In terms of MRR, KBTD outperforms all baselines except DISTMULT with an impressive improvement up to 60.4% (0.85 v.s. 0.53) on NTN. In terms of HITS@10, KBTD outperforms baselines except TransE with an improvement up to 36.9% (90.50 v.s. 66.10) on NTN. Similar results can also be observed on the FB15k dataset in Table 5. In terms of both MRR and HITS@10, KBTD outperforms all baselines except TransE with an improvement greater than 42.7% on NTN.

(2) It is notable that KBTD outperforms NTN by large margins on both datasets. We conjecture that this comes from the very high model complexity of NTN, as suggested by Table 6. In our KBTD framework, we reduce the previously considered margin-based ranking problem in the original NTN to a simple regression problem. As a result, KBTD enables the efficient training procedure to significantly boost up NTN with respect to both the MRR and HITS@10 metrics.

(3) It is also worth noting that under the KBTD framework, most models generate comparable or better results than their neural KB embedding versions. In terms of HITS@10, KBTD underperforms TransE by 9.6% (82.18 v.s. 90.90) on WN18 and 7.3% (49.94 v.s. 53.90) on FB15k. We attribute this underperformance to the constraint on TransE's scoring function. As showed in Table 3, when instantiating the KBTD framework with TransE, the scoring function is set to be the negative dissimilarity function—$f(s, r, o) = -\|\boldsymbol{a}_s + \boldsymbol{w}_r - \boldsymbol{a}_o\|_2^2$, which is for sure non-positive. However, the tensor $\mathcal{Y}$ in Eq. 7 that KBTD aims to fit allows both positive and negative entries. In specific, for the observed triplets and a moderate $b$, tensor entries are usually positive; for the corrupted triplets and a small $\epsilon$, tensor entries reach negative values. On the contrary, the scoring functions of NTN, Bilinear, and DISTMULT are able to model both positive and negative tensor entries. That said, the non-positive constraint of TransE's scoring function limits its ability to learn better latent KB representations in this link prediction task.

## 6 CONCLUSION

In this work, we provide a theoretical analysis of conventional neural KB embedding models and unveil the link between them and tensor-based KB embedding models. We show that the existing neural KB models can be unified into one tensor decomposition framework. We further propose the KBTD framework to directly fit the derived closed-form tensor. Our extensive experiments suggest that KBTD achieves consistent performance improvements over NTN, Bilinear, and DISTMULT under the neural KB embedding framework.

For further work, one interesting direction is to exploit efficient and scalable algorithms that extend KBTD to web-scale KBs. Another direction is to leverage the effective techniques from the matrix factorization community to enhance our tensor framework, such as the usage of bias terms and rich contextual information.

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
