# OpenReview forum: "Revisiting Knowledge Base Embedding as Tensor Decomposition"
_ICLR.cc/2018/Conference — Reject_

### Official Review · AnonReviewer1 · 2017-11-28

**Rating:** 3
**Confidence:** 4

**Review:**

The paper proposes a unified view of multiple methods for learning knowledge base embeddings.

The paper's motivations are interesting but the execution does fit standard for a publication at ICLR.
Main reasons:
* Section 3 does not bring much value. It is a rewriting trick that many knew but never thought of publishing
* Section 4.1 is either incorrect or clearly misleading. What happens to the summation terms related to the negative samples (o~=o' and s!=s') between the last equation and the 2 before that (on the expectations) at the bottom of page 4? They vanished while they are depending on the single triple (s, r, o), no?
* The independence assumption at the top of page 5 is indeed clearly too strong in the case of multi-relational graphs, where triples are all interconnected.
* In 4.2, writing that both RESCAL and KBTD explain a RDF triple through a similar latent form is not an observation that could explain intrinsic similarities between the methods but the direct consequence of the deliberate choice made for f(.) at the line before.
* The experiments are hard to use to validate the model because they are based on really outdated baselines. Most methods in Table 4 and 5 are performing well under their best known performance.

---

### Official Review · AnonReviewer3 · 2017-11-30

**Rating:** 5
**Confidence:** 4

**Review:**

This paper deals with the problem of representation learning from knowledge bases (KB), given in form of subject-relationship-object triplets. The paper has two main contributions: (1) Showing that two commonly used loss functions, margin-based and negative sampling-based, are closely related to each other; and (2) many of the KB embedding approaches can be reduced to a tensor decomposition problem where the entries in the tensor are a certain transformation of the original triplets values.

Contribution (1) related to the connection between margin-based and negative sampling-based loss functions is sort of obvious in hindsight and I am not sure if it has been not recognized in prior work (I'm not very well-versed in this area). Regardless, even though this connection  is moderately interesting, I am not sure of its practical usefulness. I would like the authors to comment on this aspect.

Contribution (2) that shows that KB embedding approaches based on some of the popularly used loss functions such as margin-based or negative sampling can be cast as tensor factorization of a certain transformation of the original data is also interesting. However, similar connections have been studied for word-embedding methods. For example, prior work has shown that word embedding methods that optimize loss functions such as negative sampling can be seen as doing implicit matrix factorization of a transformed version of the word-counts. Therefore contribution (2) seems similar in spirit to this line of work.

Overall, the paper does have some interesting insights but it is unclear if these insights are non-trivial/surprising, and are of that much practical utility. I would like to authors to respond to these concerns.

---

### Official Review · AnonReviewer2 · 2017-11-30

**Rating:** 3
**Confidence:** 4

**Review:**

The paper proposes a new method to train knowledge base embeddings using a least-squares loss. For this purpose, the paper introduces a reweighting scheme of the entries in the original adjacency tensor. The reweighting is derived from an analysis of the cross-entropy loss. In addition, the paper discusses the connections of the margin and cross-entropy loss and evaluates the proposed method on WN18 and FB15k.

 The paper tackles an interesting problem, as learning from knowledge bases via embedding methods has become increasingly important for tasks such as question answering. Providing additional insight into current methods can be an important contribution to advance the state-of-the-art.

However, I'm concerned about several aspects in the current form of the paper. For instance, the derivation in Section 4 is unclear to me, as eq.4 suddenly introduces a weighted sum over expectations using the degrees of nodes. The derivation also seems to rely on a very specific negative sampling assumption (uniform sampling without checking whether the corrupted triple is a true negative). This sampling method isn't used consistently across models and also brings its own problems, e.g., see the LCWA discussion in [4]

In addition, the semantics that are introduced by the weighting scheme are not clear to me either. Using the proposed method, the probability of edges between high-degree nodes are down-weighted, since the ground-truth labels are divided by the node degrees. Since these weighted labels are then fitted using a least-squares loss, this implies that links between high-degree nodes should be less likely, which seems the opposite of what the scores should look like.

With regard to the significance of the contributions: Using a least-squares loss in combination with tensor methods is attractive because it enables ALS algorithms with closed-form updates that can be computed very fast. However, the proposed method still relies on SGD optimization. In this context, it is not clear to me why a tensor framework/least-squares loss would be preferable.

Further comments:
- The paper seems to equate "tensor method" with using a least squares loss. However, this doesn't have to be the case. For instance see [1,2] which propose Logistic and Poisson tensor factorizations, respectively.
- The distinction between tensor factorization and neural methods is unclear. Tensor factorization can be interpreted just as a particular scoring function. For instance, see [5] for a detailed discussion.
- The margin based ranking loss has been proposed earlier than in (Collobert et al, 2011). For instance see [3]
- p1: corrupted triples are not described entirely correct, typically only one of s or o is corrputed.
- Closed-form tensor in Table 1: This should be least-squares loss of f(s,p,o) and log(...)?
- p6: Adding the constant to the tensor as proposed in (Levy & Goldberg, 2014) can done while gathering the minibatch and is therefore equivalent to the proposed approach.

[1] Nickel et al: Logistic Tensor Factorization for Multi-Relational Data, 2013.
[2] Chi et al: "On tensors, sparsity, and nonnegative factorizations", 2012
[3] Collobert et al: A unified architecture for natural language processing, 2008
[4] Dong et al: Knowledge Vault: A Web-Scale Approach to Probabilistic Knowledge Fusion, 2014
[5] Nickel et al: A Review of Relational Machine Learning for Knowledge Graphs, 2016.

---

### Public Comment · (anonymous) · 2017-10-30
**Not mention other SOTA results**

As shown in papers such as ProjE: Embedding Projection for Knowledge Graph Completion, Knowledge Base Completion: Baselines Strike Back and Convolutional 2D Knowledge Graph Embeddings, your results are not state-of-the-art results on WN18 and FB15k. You should mention other high published results in your paper.
The authors in the paper Convolutional 2D Knowledge Graph Embeddings analyzed and concluded that future research on knowledge base completion should not use WN18 and FB15k anymore. You should do experiments on WN18RR and FB15k-237 datasets.

---

> ### Author Response · Authors · 2017-10-31
> **The purpose of this paper is not to beat SOTA results**
>
> Dear Readers:
>
> We really appreciate your comments.
>
> For your first suggestion, we certainly noticed there are many other methods that generated superior performance on WN18 and FB15k, and we also mentioned the ProjE paper in our related work. However, the purpose of this paper is not to design the state-of-the-art methods, and we did not propose any new scoring functions. Instead, we provided the theoretical analysis on a few popular methods, and revealed that all the mentioned methods could be framed into a tensor decomposition framework. For the experiments, we are still using the same scoring functions as proposed in the mentioned papers. The purpose of the experiments is to achieve comparable results under this tensor decomposition framework. Our framework is flexible about various scoring functions.
>
> For your second suggestion, yes, we also noticed that there is FB15k-237 dataset, and thank you for pointing us the WN18RR dataset. The reason we stick with the original FB15k and WN18 datasets is for fair comparison since most of the above methods used these two datasets in their experiments. Again, we do not aim to beat one certain score function in a certain dataset. Instead, we want to generalize the existing KB embedding models, and further help the research community understand these models. We are open to work on more datasets in our future experiments though.
>
> Hope the response above clarified your questions.

---

> > ### Public Comment · (anonymous) · 2017-11-28
> > **You chose to use old models which maybe easy to beat**
> >
> > Your baseline models are old and maybe easy to beat. So I do not know your approach is good or not when comparing with strong baselines.

---

> ### Public Comment · (anonymous) · 2017-11-02
> **On FB15k and WN18**
>
> I'm not an author but felt like responding to the first comment. I agree with the statement that the paper is missing SOTA results. With some minor tuning of TransE and DistMult one can achieve *much* better numbers than the one reported in the paper. For instance, on FB15k it is possible to get up to 90 hits@10 with DistMult and 76.x with TransE. That's clearly a weak point of the paper irrespective of the theoretical analysis which might or might not be insightful. I haven't looked at this part of the paper at all.
>
> I disagree with the comment on FB15k and WN18. First, the "problems" with FB15k and WN18 weren't first mentioned by the ConvE authors but much earlier by Toutanova et al [1]. The identified "problems" relate to the existence of reverse relations that allow simple baselines to perform better than most (at the time) existing KB embedding methods. For example, the existence of (A, parentOf, B) predicts with high probability the relation (B, childOf, A). However, I disagree with the conclusion that these two data sets should not be used anymore. There are still plenty of challenging completion queries in these data sets. Also, FB15k is a data set derived from a human-designed and populated knowledge base. It is somewhat absurd to now exclusively use artificially created KBs to evaluate KB completion methods. FB15k-237 and WN18RR are data sets that should be included in addition to FB15k and WN18.
>
>
> [1] Observed Versus Latent Features for Knowledge Base and Text Inference. Kristina Toutanova and Danqi Chen.

---

> > ### Public Comment · (anonymous) · 2017-11-28
> > **On FB15k-237 and WN18RR**
> >
> > Toutanova et al. [1] firstly showed the problem, but the authors of ConvE firstly showed SOTA results on FB15k and WN18 by using a simple reversal rule. It's fine if we don't know this. Otherwise, the question is why we use the models that are outperformed by this simple reversal rule on FB15k and WN18???
> > FB15k-237 and WN18RR should become the main datasets for the link prediction task.

---

### Decision · Program_Chairs · 2018-01-29
**ICLR 2018 Conference Acceptance Decision**

**Decision:**

Reject

**Comment:**

The reviewers are not convinced by a number of aspects: including originality and clarity. Whereas the assessment of clarity and originality may be somewhat subjective (though the connections between margin-based loss and negative sampling is indeed well known), it is pretty clear that evaluation is very questionable. This is not so much about existence of more powerful factorizations  (e.g., ConvE / HolE) but the fact that the shown baselines (e.g., DistMult) can be tuned to yield much better performance on these benchmarks.  Also, indeed the authors should report results on cleaned versions of the datasets (e.g., FB15k-237).  Overall, there is a consensus that the work is not ready for publication.

Pros:
-- In principle, new insights on standardly used methods would have been very interesting

Cons:
-- Evaluation is highly problematic
-- At least some results do not seem so novel / interesting; there are questions about the rest (e.g., assumptions)
-- The main advantage of sq loss methods is that it enables the alternating least squares algorithm, does not seem possible here (at least not shown)